# Phosphate Additives for Aging Inhibition of Impregnated Activated Carbon against Hazardous Gases

**DOI:** 10.3390/ijms241613000

**Published:** 2023-08-20

**Authors:** Ido Nir, Vladislav Shepelev, Alexander Pevzner, Daniele Marciano, Lilach Rosh, Tal Amitay-Rosen, Hadar Rotter

**Affiliations:** 1Department of Physical Chemistry, Israel Institute for Biological Research, P.O. Box 19, Ness Ziona 74100, Israel; alexanderp@iibr.gov.il (A.P.); lilachm@iibr.gov.il (L.R.); tala@iibr.gov.il (T.A.-R.); hadarr@iibr.gov.il (H.R.); 2Life Science Research Israel Ltd. (LSRI), P.O. Box 19, Ness Ziona 74100, Israel; vladislavs@iibr.gov.il; 3Department of Organic Chemistry, Israel Institute for Biological Research, P.O. Box 19, Ness Ziona 74100, Israel; daniele.marciano@gmail.com

**Keywords:** aging, activated carbon, cyanogen chloride, phosphate, impregnation, adsorption

## Abstract

Impregnated activated carbons (IACs) used in air filtration gradually lose their efficacy for the chemisorption of noxious gases when exposed to humidity due to impregnated metal deactivation. In order to stabilize IACs against aging, and to prolong the filters’ shelf life, inorganic phosphate compounds (phosphoric acid and its three salts, NaHPO_4_, Na_2_HPO_4_, and Na_3_PO_4_) were used as anti-aging additives for two different chromium-free IACs impregnated with copper, zinc, molybdenum, and triethylenediamine (TEDA). Phosphoric acid, monosodium, and disodium phosphate were found to be very efficient in inhibiting the aging of IACs over long periods against cyanogen chloride (the test agent) chemisorption, with the latter being the most efficient. However, the efficiency of phosphate as an anti-aging additive was not well correlated with its ability to inhibit the migration of metal impregnants, especially copper, from the interior to the external surface of carbon granules. Unlike organic additives, the inorganic phosphate additives did not decrease the surface area of the IAC or its physical adsorption capacity for toluene. Using a phosphate additive in IAC used in collective protection and personal filters can improve the safety of the user and the environment and dramatically reduce the need to replace these filters after exposure to humid environments. This has safety, economic, logistical, and environmental advantages.

## 1. Introduction

Activated carbon (AC) is a form of carbon that has been processed to have a well-developed porous structure, a high surface area, and a high micropore volume available for adsorption. Due to its high degree of microporosity, one gram of activated carbon may have a surface area between 1000 and 2000 m^2^ or even greater [1].

AC has many commercial uses, including the filtration of noxious gases from the atmosphere. Nonimpregnated AC is generally effective in adsorbing organic compounds that do not exhibit high volatility (i.e., boiling points greater than 65 °C). However, for effective filtration of volatile organic compounds and low-molecular-weight gases, such as hydrogen cyanide (HCN), cyanogen chloride (denoted as CK or CNCl), sulfur dioxide, ammonia, formaldehyde, and chlorine gas, it is necessary to impregnate AC with metal additives [2,3,4,5,6,7,8,9,10]. The target gases adsorb more effectively to the impregnated metals through chemical adsorption than to the AC via physical adsorption [2,6]. Chromium, together with copper and a small amount of silver, was initially found to be an effective additive combination. However, the use of chromium was discontinued after it was discovered to be a carcinogen [5]. Commonly used additives today include salts of copper, zinc, molybdenum, silver, and triethylenediamine (TEDA). One of the most widespread commercially produced forms of impregnated activated carbon (IAC) with such additives is ASZM-TEDA^®^ activated carbon (denoted as ASZMT), produced by Calgon Carbon Corporation [6]. ASZMT is effective for removing gases and/or vapors, including vapors of organic compounds, chlorine, hydrogen chloride, hydrogen cyanide, cyanogen chloride, sulfur dioxide, hydrogen sulfide, formaldehyde, and others [6].

It is well known that IACs gradually lose their efficacy when exposed to humidity. Adsorbed water allows the dissolution of impregnated metals and their migration to the external surface of the carbon granules [11,12,13] and causes the size of the metal crystals to increase [14,15].

As a result, the metal impregnants gradually lose their efficacy against certain noxious gases, especially the blood chemical warfare agents HCN and CK [4,6,12].

Using XPS and EDS methods, Rossin and Morison [12] showed that exposure of ASZ carbon (which contains copper, silver, and zinc only) to humidity and aging results in copper leaching from within the granules to the external surface, while less zinc migrates to the external surface. Following the aging of the IAC, exposure to HCN led to the premature breakthrough of cyanogen, which is a reaction product of HCN and divalent copper [16]. The premature cyanogen breakthrough was attributed to the migration of divalent copper to the external surface of the carbon granules. The addition of molybdenum and TEDA to ASZ carbon (ASZMT) reduces copper migration, thereby improving the performance of the aged carbon against HCN [11]. However, as seen in the current study, the performance of ASZMT against the chemically adsorbed warfare agent CK deteriorates significantly with aging.

Aging of collective protection and personal-use filters has safety, economic, logistical, and environmental consequences. Filters in use, or filters in storage, that have been exposed to moisture may become ineffective, sometimes without the user’s knowledge. Replacing them involves high economic costs and presents a logistical burden. In addition, the transportation and burial of discarded filters at waste sites have environmental consequences.

Accordingly, it would be advantageous to devise an additive to IAC that prolongs its shelf life by reducing its sensitivity to moisture exposure and aging.

Despite the serious consequences of aging, the amount of research that has been conducted to solve this problem is limited. However, several studies have been published over the years in which additives were introduced to carbon to improve its initial protection value or inhibit its aging rate.

Hittanau et al. [17] showed that after 7 days of accelerated aging at 46 °C and a relative humidity of 80%, activated carbon impregnated with copper, zinc, molybdenum, TEDA, sulfate, and 0.6% phosphate had a slightly higher breakthrough time for CK than the same carbon with 0.4% phosphate (42 and 31 min, respectively). The breakthrough time for carbon that did not contain the additive was not shown. In several studies, organic amines have been used to improve the efficiency of AC and IAC against CK [4,18,19,20,21]. Organic compounds can form a complex with metal additives, thereby improving their stability and chemisorption efficiency. Pyridine carboxylic acid (PCA 4), added to nonimpregnated AC, chemically reacts with CK and increases adsorption efficiency [18]. It was found that AC containing the PCA 4 additive was not sensitive to aging. Deitz and Karwacki showed that with the addition of 3 wt.% acetoacetamide (acac) to ASZMT, the initial breakthrough time (without aging) for CK increased from 28 to 44 min, and they proposed a mechanism for the role of acac in the reaction [19]. In a further study [20], they proposed a model for Whetlerite IACs in which adsorbed copper ions and selected co-adsorbed ligands (several organic compound additives such as amines) influenced the electron transfer from cyanogen vapor to the metal ions. The ligand species were considered to form a coordination complex with the copper ions and the cyano group in the feed stream. The two complexes differed only slightly regarding free energy values and could transfer electrons from the cyano groups to the metal ions.

TEDA is considered the most successful organic amine additive and was found to improve the efficiency of ASC carbon (containing Cu, Cr, and Ag impregnants) against CK under humid conditions. Moreover, it considerably reduced the effect of aging [4,22]. It is commonly assumed that TEDA shields the copper–chromium complex from degradation by adsorbed water [4]. Notably, TEDA plays a crucial role in the performance of ASZMT against CK. It does not only act as an anti-aging additive but also dramatically improves the protection capacity of the new ASZMT against CK [6,23]. Mahle et al. [23] attributed the effect of TEDA to the formation of soluble complexes between Cu and TEDA.

The aim of the present work was to develop a chromium-free IAC containing phosphate additives to improve the material’s resistance to aging over long periods. Phosphate compounds can coordinate with transition metals [24,25,26,27], such as copper and zinc impregnants, and possibly improve the IAC’s stability against the influence of high humidity. Unlike organic additives, inorganic water-soluble phosphate additives do not occupy physical adsorption sites. Thus, no reduction in the IAC’s protection value against physically adsorbed compounds is expected. For this purpose, AC was impregnated with copper, zinc, molybdenum, and TEDA, similar to commercially available ASZMT. The effect of phosphate addition on the protection efficiency of laboratory-impregnated carbon and commercial ASZMT after long accelerated aging periods (up to 6 months) was investigated. The influence of phosphate type (phosphoric acid and its three sodium salts) and concentration are discussed. As the protection capacity of the filter material against CK significantly decreases during aging, CK was used as the testing agent.

## 2. Results and Discussion

### 2.1. CK Adsorption on IAC

#### 2.1.1. Pre-Humidified IAC

Pre-humidified ASZMT and LIAC were exposed to CK gas before and after accelerated aging until a breakthrough occurred. Figure 1 shows the results of the breakthrough curve measurements, presented as protection values (Ct, g min/m^3^), which are calculated as the product of the breakthrough time and the influent concentration (C_0_ × t_B_).

For both ASZMT and LIAC, the protection values against CK dramatically decreased after 90 days of accelerated aging, rendering them virtually ineffective. The results in Figure 1 emphasize the benefits of inhibiting the aging rate.

#### 2.1.2. Effect of Phosphate Additive (NaH_2_PO_4_·H_2_O) on IAC Aging

As a potential anti-aging additive, sodium phosphate monobasic monohydrate (NaH_2_PO_4_·H_2_O) was added to the IACs. Figure 2A,B show the effect of this phosphate additive on the Ct values against CK before and after aging for ASZMT (2A) and LIAC (2B) IACs.

As shown in Figure 2, both phosphate-modified IACs presented only a 20–25% decrease in protection values against CK after 3 months of accelerated aging, unlike 84–96% without the additive.

For both IACs, the addition of phosphate led to a reduction in the initial Ct value of the new carbon (before aging), from 170 to 120 g min/m^3^ for the ASZMT and from 171 to 148 g min/m^3^ for the LIAC. Usually, in commercial filters, the activated carbon content exceeds the minimum amount needed for protection against CK. Therefore, the reduction in the initial Ct value is not necessarily significant. The increase in the carbon shelf life far outweighs the loss in the initial protection value.

The effect of drying on the protection values of aged IACs is shown in Figure 3. All carbons (new and aged) presented in Figure 3 were dried at 80 °C before the breakthrough measurements.

The results demonstrate that, similar to humidified carbon, accelerated aging for 3 and 6 months reduced the efficacy of dried pristine IACs (ASZMT and LIAC) against CK by 59–89% and >75%, respectively. The addition of phosphate inhibited the aging effect, and the protection value decreased by only 17–23% and 32–37% after 3–6 months of aging. In general, pre-drying the aged carbon increased the protection values compared to humidified carbon.

The reason for this is that after aging, particularly in IACs without an additive, the capacity of the impregnated metals for chemical adsorption was very low. Therefore, CK adsorption occurred mainly via physical adsorption. Drying removed water occupying physical adsorption sites, enabling some amount of CK adsorption through physical adsorption. In practice, even when IACs have aged and are past their expected shelf life, their protection capability may be partially recovered by drying.

#### 2.1.3. Effect of Phosphate Concentration (NaH_2_PO_4_·H_2_O) on IAC Aging

To evaluate the optimum phosphate concentration, further experiments were conducted by varying the concentration of NaH_2_PO_4_·H_2_O in the IACs. The results are shown in Figure 4.

As seen in Figure 4A, a concentration of 3.38 wt.% phosphate additive was sufficient to achieve the minimum reduction in the protection values of aged ASZMT. For the LIAC, the maximum efficiency was obtained with 5.15 wt.% phosphate additive.

#### 2.1.4. Effect of Phosphate Type on LIAC Aging

To evaluate whether the phosphate additive type influences the additive’s efficacy in preventing LIAC aging, further experiments were performed with other phosphate compounds (H_3_PO_4_, Na_2_HPO_4_·2H_2_O, and Na_3_PO_4_·12H_2_O). The effects of phosphoric acid (H_3_PO_4_) and phosphate dibasic dihydrate (Na_2_HPO_4_·2H_2_O) on the protection values of aged LIAC against CK are presented in Figure 5 and Figure 6, respectively.

As seen, phosphoric acid at a concentration of 5.15 wt.% was highly effective in inhibiting aging (the protection values remained unchanged after 6 months of accelerated aging). However, the initial protection value (for a new carbon) was significantly reduced with 5.15 wt.% phosphoric acid addition, from 171 to 95 g min/m^3^. Overall, phosphoric acid was a less favorable additive than NaH_2_PO_4_.

Experiments using the Na_2_HPO_4_·2H_2_O additive were conducted at a concentration of 6.52 wt.%, the molar equivalent of a NaH_2_PO_4_·H_2_O concentration of 5.15 wt.%. The results in Figure 6 demonstrate that the Na_2_HPO_4_·2H_2_O additive was also effective in preventing aging but was not as effective as NaH_2_PO_4_·H_2_O. For example, after aging for 180 days, the protection value of LIAC with the Na_2_HPO_4_·2H_2_O additive declined to 65 mg min/m^3^ compared to 107 mg min/m^3^ for LIAC with the NaH_2_PO_4_·H_2_O additive.

Similar experiments were performed with trisodium phosphate 12 hydrate (Na_3_PO_4_·12H_2_O) on LIAC, which was dried after aging. The experiments were conducted at a concentration of 11.03 wt.%, the molar equivalent of NaH_2_PO_4_·H_2_O at 4.00 wt.%. It was evident that trisodium phosphate was not efficient in inhibiting aging. After 3 months of accelerated aging, the protection values were even lower than those for unmodified LIAC. However, it is possible that the efficiency of Na_3_PO_4_·12H_2_O would have been better if it was added in an acidic environment or with an acid or acidic compound. This is because acidic conditions may contribute a proton to the phosphate, causing the sodium phosphate to function effectively as Na_2_HPO_4_ or NaH_2_PO_4_.

Overall, the results suggest that the monobasic salt NaH_2_PO_4_·H_2_O was the most effective additive in inhibiting aging among the phosphate additives consisting of phosphoric acid and its three sodium salts.

### 2.2. The Effect of Phosphate on the Physical Adsorption of Toluene

The addition of additives to activated carbon can lead to a decrease in physical adsorption capacity due to a loss of adsorption sites or partial micropore blockage. Toluene adsorption measurements were conducted to ensure that the addition of phosphate additives did not reduce the physical adsorption capacity of IACs, i.e., did not decrease the protection values against physically adsorbed organic compounds. Unlike cyanogen chloride, toluene is physically adsorbed onto activated carbon. The experimental results presented in Table 1 demonstrate that adding NaH_2_PO_4_·H_2_O to ASZMT or LIAC did not decrease the protection values against toluene. These results indicate the advantage of using phosphate as an additive to prevent IAC aging compared to organic additives. Unlike inorganic phosphate, organic additives are physically adsorbed and occupy adsorption sites, leading to a reduction in the protection values against other target substances by competitive physical adsorption [4].

### 2.3. Effect of a Phosphate Additive on the Textural Properties of IACs

The textural properties of the IACs, obtained from N_2_ adsorption–desorption isotherms, were determined to study the effect of additive addition (NaH_2_PO_4_·H_2_O) on processes such as micropore blockage. The N2 adsorption–desorption isotherms and the textural properties are presented in Figure 7 and Table 2.

The N_2_ adsorption–desorption isotherms in Figure 7 are typical for a microporous material (type I) like microporous activated carbon [1]. The phosphate addition did not change the isotherm shape, and no indication of micropore blockage was observed.

Table 2 shows that the addition of NaH_2_PO_4_·H_2_O did not cause a significant reduction in the surface area and micropore volume of the activated carbon samples. This is consistent with the results obtained for toluene adsorption.

### 2.4. Effect of a Phosphate Additive on Impregnated Metal Migration to the External Carbon Surface

Migration of impregnated metals to the external surface of IAC granules is suspected to be a significant aging mechanism leading to the loss of the IAC’s protection efficacy against certain noxious gases, especially HCN and CK [11,12,13]. To investigate the effect of a phosphate additive on the IAC metal migration mechanisms, EDX elemental analysis was conducted, and the results are presented in Table 3.

In both new IACs, the copper concentration, and particularly the zinc concentration on the external surface of the carbon granules, were higher than their concentrations in the bulk AC (5 wt.% Cu and 5 wt.% Zn). This implies that the distribution of metals, especially zinc after the impregnation process, was not uniform. These results are consistent with previous studies [11,28] and may be attributed to zinc migration from the bulk to the external surface during drying after metal impregnation.

During aging, migration of copper and zinc to the external surface occurred in the ASZMT. After 3 months of accelerated aging, the copper and zinc concentrations increased from 8.4 and 23.8 wt.% to 23.4% and 37.5 wt.t%, respectively. Most of the metal migration occurred during the first 3 months of accelerated aging.

Adding the phosphate additive NaH_2_PO_4_·H_2_O to the commercial ASZMT decreased the zinc concentration on the external surface of the new carbon. This is probably due to the entrainment of zinc from the external surface to the interior during the introduction and absorption of the phosphate solution on the carbon. In phosphate-modified LIAC, phosphate is added to carbon before metal impregnation. This explains why the metal concentrations in the new phosphate-modified LIAC are not significantly lower than those in the new LIAC without phosphate.

Interestingly, Table 3 shows an important finding regarding LIAC. Although phosphate addition was very effective in preserving the carbon efficiency against CK during aging, it did not inhibit copper migration to the external surface. The copper concentration on the external surface of LIAC with phosphate increased from 13.3 wt. % to 21.8 and 25.3 after 3 and 6 months, respectively. Therefore, it can be concluded that the migration of the copper to the external surface of the carbon is not necessarily the dominant mechanism for the deterioration in the protection value against CK. In contrast, in ASZMT, the efficiency of aging inhibition was lower (Figure 2), although phosphate efficiently inhibited copper migration. After 3 months of accelerated aging, the copper concentration increased to 23.2 wt. % in the unmodified ASZMT, compared to only 8.8 wt.% in the phosphate-modified ASZMT. This supports the finding that the migration of copper is not necessarily the dominant mechanism for the deterioration in protection values against CK during aging. In the previous literature [11,12], it was hypothesized that the migration of copper to the external surface is responsible for the premature cyanogen breakthrough upon exposure to HCN. This does not contradict the findings in the current work since the adsorption mechanisms for CK and HCN are different. It should be kept in mind that since the external surface area of carbon is much lower than its total surface area, the amount of metal that accumulates on the external surface of IACs is probably just a small fraction of the total metal content in the carbon. This could explain why the protection values against CK of the modified LIAC remained relatively high after aging, despite copper migration.

## 3. Materials and Methods 

### 3.1. Chemicals

Coal-based nonimpregnated AC (BPL) with a 12 × 30 mesh size (Calgon Carbon Corporation, Pittsburg, PA, USA) was used as the substrate for the impregnated carbons. This carbon is denoted in the text as BPL. Coal-based impregnated AC (ASZMT), 12 × 30 mesh size (Calgon Carbon Corporation, Pittsburg, PA), denoted in the text as ASZMT, was used for the breakthrough measurements. This carbon contained impregnants of copper (5 wt.%), zinc (5 wt.%), molybdenum (1.8 wt.%), silver (0.05 wt.%), and TEDA (3 wt.%) on a BPL substrate.

Zinc carbonate basic (97%, Alfa Aesar, Ward Hill, MA, USA), copper (II) carbonate basic (98% Sterm Chemicals, Newburyport, MA, USA), and ammonium molybdate tetrahydrate (99%, Thermo Fisher Scientific, Waltham, MA, USA) were used as the sources for metal impregnation. Ammonium hydroxide solution (25%, Bio Lab, Jerusalem, Israel) and ammonium carbonate (30% NH_3_, Alfa Aeser) were used for the preparation of the impregnation solution. Phosphoric acid (85% in water, Sigma, Rehovot, Israel), sodium phosphate monobasic monohydrate (98%, Glentham, Corsham, UK), sodium phosphate dibasic dihydrate (98.5%, Sigma), and sodium triphosphate 12-hydrate (99%, Merck, Darmstadt, Germany) were used as phosphate additives for the carbon, and TEDA (98%, Glentham) was used as the amine additive for the carbon. A cyanogen chloride (CK) cylinder with a purity greater than 98.5% was purchased from Liquidgas LTD, Israel. Toluene (99.5%, Merck) was used for breakthrough measurements.

### 3.2. Impregnated Activated Carbon Preparation

The BPL AC was impregnated in our laboratory with 5 wt.% copper, 5 wt.% zinc, 2 wt.% molybdenum, and 3 wt.% TEDA. This IAC is denoted in the text as LIAC (laboratory-impregnated activated carbon). The LIAC was similar but not identical to ASZMT. Specifically, while ASZMT contains 0.05% silver, LIAC does not include this metal.

The BPL was impregnated in two consecutive cycles using the incipient wetness method, in which the impregnation solution was slowly added to the substrate while stirring until the activated carbon started to stick to the glass walls. An aqueous impregnated solution was prepared, comprising salts of copper carbonate basic, zinc carbonate basic, and ammonium molybdate, according to the desired metal content in the final LIAC product. The impregnation solution also contained ammonia, ammonium hydroxide, and ammonium carbonate to aid the dissolution of the copper and zinc salts in the aqueous solution [6]. The impregnation solution was dripped into the AC at a volume of 0.7 mL for 1 g of AC, just slightly below the wetness volume. The AC was dried at 130 °C for at least 30 min. In the second cycle, the impregnation solution was dripped into the activated carbon at a volume of 0.55 mL/g. The AC was dried at 100 °C for 30 min, 130 °C for 30 min, 160 °C for 45 min, and 180 °C for 45 min. The water, ammonia, and ammonium carbonate evaporated during this drying process. The copper and zinc ions precipitated in the pores of the activated carbon, and due to the high temperature, they were converted, at least partially, to zinc oxide (ZnO) and cupric oxide (CuO) [6,29,30], both of which have extremely low solubility in water. After drying, TEDA was added to the activated carbon via incipient wetness impregnation at a volume of 0.43 mL/g with mixing. The quantity of the added TEDA was 3 wt.% based on the final weight of the carbon. The carbon was dried again at 80 °C, a temperature that is low enough to ensure negligible TEDA evaporation from the carbon.

### 3.3. Addition of Phosphate Additives

The addition of phosphate additives to the activated carbons was performed using the incipient wetness impregnation method, as described above. Four phosphate compounds were used, namely, phosphoric acid and its three salts (monobasic, dibasic, and tribasic). A phosphate aqueous solution with the necessary amount of phosphate compound (0.6–11 wt.% based on the final carbon weight) was dripped onto BPL and ASZMT at volumes of 0.7 g/mL and 0.43 g/mL, respectively. The final products were dried at 80 °C for 3 h. The phosphate-modified BPL was then impregnated with metal salts and TEDA, as described in Section 2.2, to obtain the phosphate-modified LIAC.

In the text, the phosphate concentration in all IACs is expressed as the weight of the whole phosphate compound based on the final carbon weight (including the weight of the impregnants and additives).

### 3.4. Preparation of Activated Carbon Beds

Sixty-millimeter diameter beds were prepared in our laboratory by filling anodized aluminum columns with 33.8 g ASZMT or LIAC (based on the IAC weight without phosphate) using the “snowfall” technique [31]. The column bed was retained between round discs made of rigid, perforated fiberglass placed at the top and bottom of the column and covered with filter paper to prevent carbon dust from passing through. The bed was compacted by pressing and tightening an annular screw mounted on top of the upper retaining disc to a constant torque of 7 N m. Both carbon types had a bulk density of 0.63 g/cm^3^ and a bed depth of 1.85 cm.

### 3.5. IAC Pretreatments

Prior to the adsorption experiments, most IACs were subjected to the following treatments.

#### 3.5.1. Humidification of the Carbon Beds

The carbon beds were pre-humidified with a 20 L/min airflow at a temperature of 18.0 ± 0.2 °C and a relative humidity (RH) of 85 ± 1%. The RH of the airflow was monitored using a humidity sensor (HyCal Model 829, El Monte, CA, USA). Equilibrium was assumed to be reached when the bed weight remained within 0.02 g between successive weighings at 2 h intervals. Both the commercial ASZMT and LIAC achieved a constant water content of 31.0 ± 1% by weight (per dry carbon weight) in less than 48 h.

#### 3.5.2. Accelerated Aging

After pre-humidification, some carbon beds were sealed and stored in a thermostatic chamber at 50 ± 0.5 °C. The aging periods were 1.5, 3, or 6 months.

#### 3.5.3. Carbon Bed Drying

Some of the carbon beds were pre-dried at 80 °C for 3 h. No further water loss was detected for longer times (<0.02 g). After drying, the carbon beds had a water content of approximately 1%.

### 3.6. CK Dynamic Adsorption

Dynamic breakthrough experiments for CK adsorption on carbon beds were performed before and after aging. The AC beds were subjected to 940 ppm (2.4 mg/L) CK (generated from a CK cylinder) at a flow rate of 10 L/min (linear flow velocity of 5.9 cm/s) at a temperature of 18 ℃ and relative humidity of 75% RH. These flow rates are equivalent to approximately 30 L/min through a typical personal gas-mask filter with a 105 mm carbon bed diameter.

The breakthrough time, t_B_, was defined as the time when the CK effluent concentration reached 0.59 ppm (1.5 μg/L, C_0_/C_x_~600). The effluent concentration was determined using a colorimetric method in which a solution of pyridine/water/dimedone (at a ratio of 90 mL/10 mL/1 g) changed color from transparent to pink due to a reaction with CK. The validity of using the color change in solution as an indicator at a CK concentration of 0.59 ppm was confirmed using a GC (model 8610C, SRI Instruments) instrument equipped with a calibrated DELCD (Dry Electronic Conductivity Detector).

### 3.7. Toluene Dynamic Adsorption

The dynamic breakthrough experiments for toluene adsorption on the carbon beds were performed at a toluene concentration of 1000 ppm (3.5 mg/L), a flow rate of 15 L/min (linear flow velocity of 8.8 cm/s), a temperature of 40 °C, and a relative humidity of %RH < 20%. This flow rate is equivalent to approximately 45 L/min through a typical personal gas-mask filter with a 105 mm carbon bed diameter.

The effluent concentration of toluene was continuously monitored with a calibrated photoionization detector (model ppbRAE 3000, RAE systems, San Jose, CA, USA). The breakthrough time, t_B_, was defined as the time in which the toluene effluent concentration reached 0.35 μg/L (100 ppb, C_0_/C_x_~10,000).

### 3.8. Carbon Characterization

Several methods were used to characterize the changes in the metal impregnants and carbon surface after aging:I.Changes in the surface elemental composition of the carbon adsorbents were analyzed using energy-dispersive X-ray spectrometry (EDX) using a PhenomProX scanning electron microscope (SEM) produced by Thermo Fisher Scientific. All data were recorded using a 15 keV electron acceleration voltage. Since the metal distribution on the external surface of the carbon is heterogeneous even for a given granule, the elemental concentration was determined as the average of 10 different areas on different granules. The analyzed area in each measurement was 316 μm × 316 μm (magnification of 850×).II.The Brunauer‒Emmett‒Teller (BET) surface area and pore volume were obtained from N_2_ adsorption–desorption isotherms at 77 K using a NOVA 1200 e (Quantachrome) system. Before analysis, the samples were outgassed under vacuum at 80 °C. The micropore surface area was derived from a t-plot analysis of the adsorption isotherm. The outgassing was performed at a temperature of 80 °C because, at higher temperatures, a significant sublimation of TEDA from the IAC may occur. Furthermore, the sublimation rate of the TEDA from the IAC at high temperatures may vary for the different types of IACs.

## 4. Conclusions

In the current study, we developed a chromium-free IAC containing phosphate additives to improve the filter material’s resistance to aging over long periods. The addition of phosphate, particularly the monobasic phosphate NaH_2_PO_4_, to AC impregnated with copper, zinc, molybdenum, and TEDA (the most common IAC in the USA and Europe) was found to be highly effective in inhibiting aging effects against CK. The protection value against CK decreased by only ~20% after 3 months of accelerated aging compared to 96% without the additive. Unlike organic additives, the phosphate additives do not occupy physical adsorption sites, and therefore, no reduction in the protection value against a physically adsorbed compound (toluene) was observed. The efficiency of the phosphate additive in LIAC was very high, although it did not inhibit the migration of the copper from the interior to the external surface of the carbon granules. This finding suggests that the migration of metals to the external surface of carbon is not necessarily the dominant mechanism for the deterioration in the protection values against chemically adsorbed gases.

Inhibiting IAC aging in collective protection and personal-use filters has positive safety, economic, logistical, and environmental consequences.

## Figures and Tables

**Figure 1 ijms-24-13000-f001:**
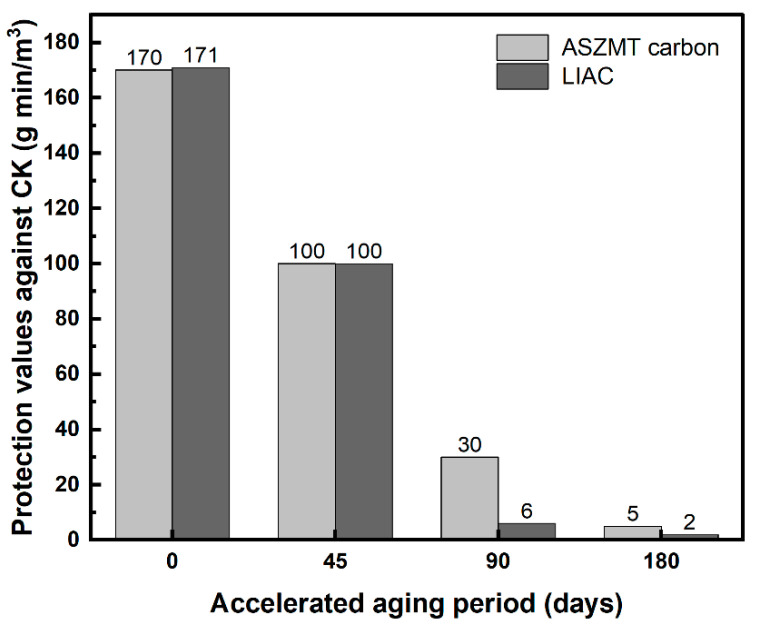
Effect of aging on commercial ASZMT and LIAC protection values against CK.

**Figure 2 ijms-24-13000-f002:**
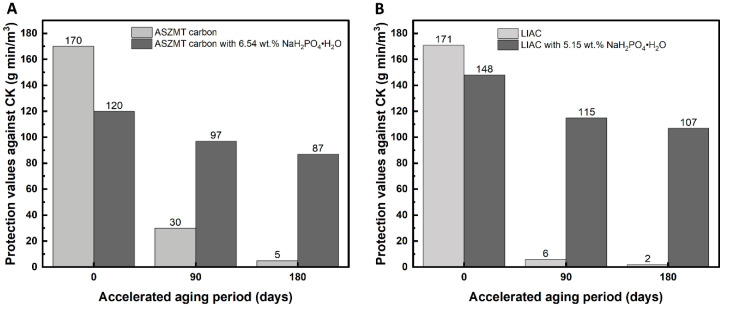
Effect of phosphate additive (NaH_2_PO_4_·H_2_O) on CK protection values of aged impregnated activated carbon; (**A**) ASZMT, (**B**) LIAC.

**Figure 3 ijms-24-13000-f003:**
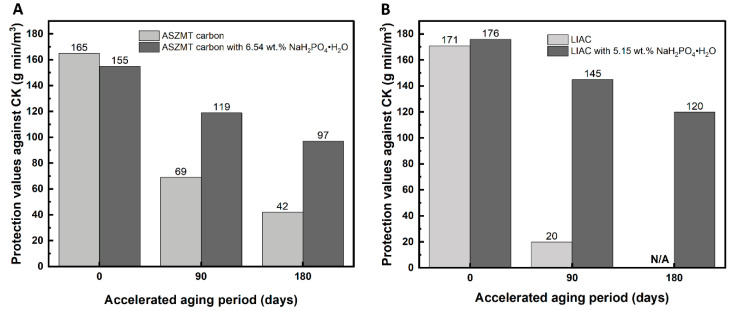
Effect of phosphate additive (NaH_2_PO_4_·H_2_O) on CK protection values of dried, aged, impregnated activated carbon; (**A**) ASZMT, (**B**) LIAC.

**Figure 4 ijms-24-13000-f004:**
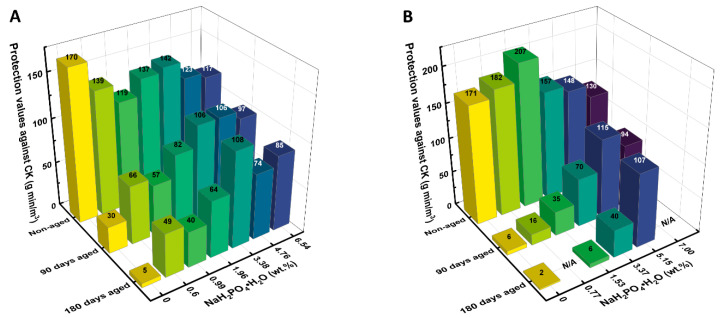
Effect of phosphate additive (NaH_2_PO_4_·H_2_O) concentration on the protection values of new and aged, humidified, impregnated activated carbons; (**A**) ASZMT, (**B**) LIAC.

**Figure 5 ijms-24-13000-f005:**
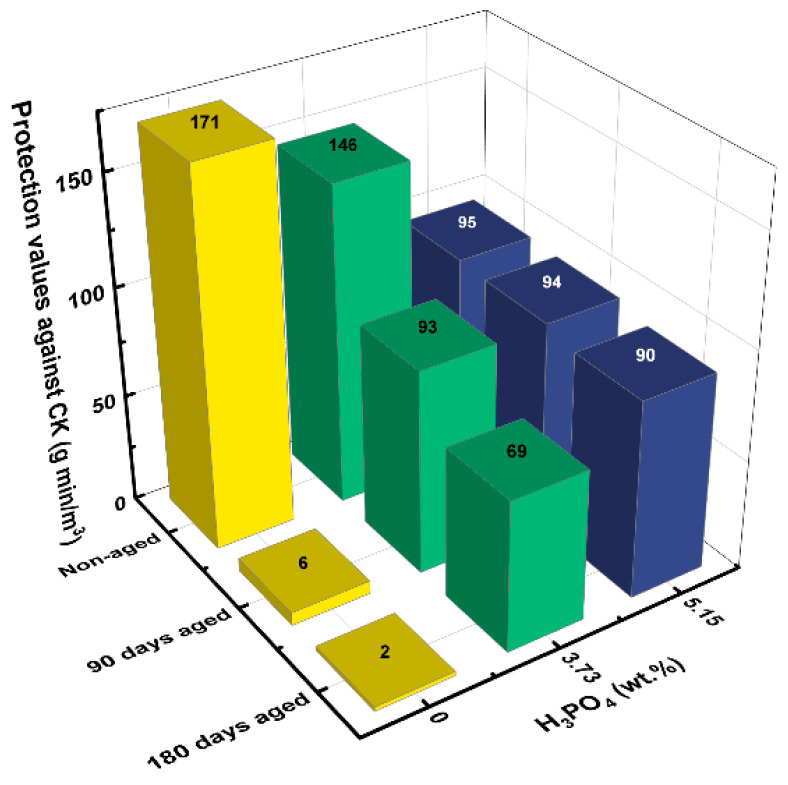
Effect of phosphoric acid concentration on new and aged, humidified LIAC protection values.

**Figure 6 ijms-24-13000-f006:**
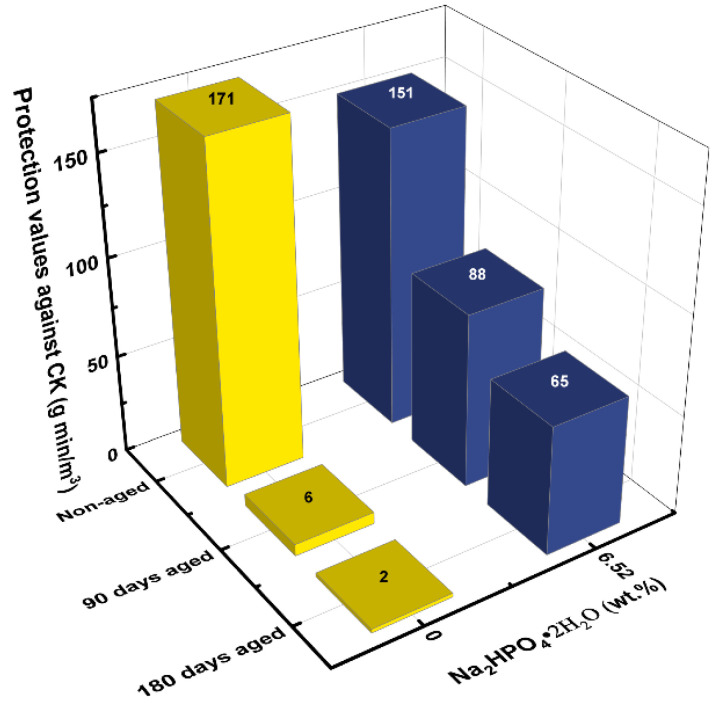
Effect of phosphate dibasic dihydrate (Na_2_HPO_4_·2H_2_O) on the protection values of new and aged, humidified LIAC.

**Figure 7 ijms-24-13000-f007:**
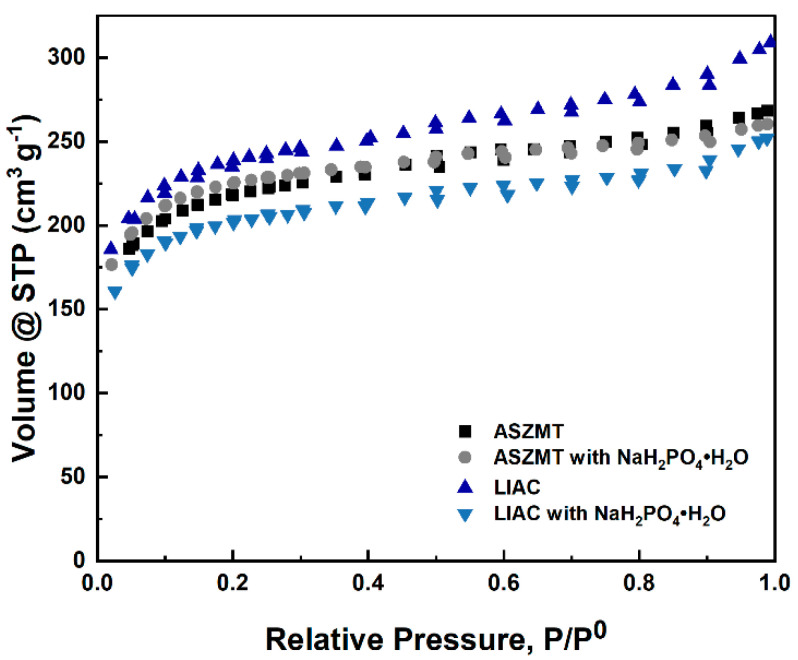
N_2_ adsorption-desorption isotherms of IACs.

**Table 1 ijms-24-13000-t001:** The effect of a phosphate additive on the protection values (g min/m^3^) against toluene.

ASZMT without Phosphate	ASZMT with 6.54 wt.% NaH_2_PO_4_·H_2_O	LIAC without Phosphate	LIAC with 5.15 wt.% NaH_2_PO_4_·H_2_O
347	394	331	356

**Table 2 ijms-24-13000-t002:** Effect of a phosphate additive on the textural properties of IACs.

	ASZMT without Phosphate	ASZMT with 6.54 wt.% NaH_2_PO_4_·H_2_O	LIAC without Phosphate	LIAC with 5.15 wt.% NaH_2_PO_4_·H_2_O
Surface area (m^2^/g) (BET)	687	699	734	696
% microporosity	80.1	84.0	81.1	81.1
Micropore volume (cm^3^/g)	0.295	0.305	0.298	0.288

**Table 3 ijms-24-13000-t003:** Weight concentration (% wt.) of Cu, Zn, and Mo on the external surface of new and aged IAC granules (EDX analysis).

IAC Type	Cu	Zn	Mo
**ASZMT–new**	8.4	23.8	3.7
**ASZMT–aged 3 M**	23.4	37.5	4.2
**ASZMT–aged 6 M**	23.2	40.7	2.9
**ASZMT with 6.54 wt.% NaH_2_PO_4_·H_2_O–new**	6.3	13.1	2.8
**ASZMT with 6.54% NaH_2_PO_4_·H_2_O–aged 3 M**	8.9	15.6	3.0
**ASZMT with 6.54% NaH_2_PO_4_·H_2_O–aged 6 M**	8.8	17.5	2.6
**LIAC–new**	8.9	28.5	2.5
**LIAC–aged 3 M**	20.8	30.9	5.2
**LIAC–aged 6 M**	19.6	26.5	5.1
**LIAC with 5.15 wt.% NaH_2_PO_4_·H_2_O–new**	13.3	23.5	1.7
**LIAC with 5.15% NaH_2_PO_4_·H_2_O–aged 3 M**	21.8	22.3	2.6
**LIAC with 5.15% NaH_2_PO_4_·H_2_O–aged 6 M**	25.3	28.5	2.1

## Data Availability

Data is contained within the article.

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
