# Peer review of "Phosphate Additives for Aging Inhibition of Impregnated Activated Carbon against Hazardous Gases"

_ijms, 2023, doi:10.3390/ijms241613000_

Round 1
Reviewer 1 Report
The paper entitled “Phosphate additives for aging inhibition of impregnated activated carbon against hazardous gases” by Nir et al. is interesting study and authors have done good job. However, here are few comments which need to be answered:
1. The authors are advised to check the grammar of the manuscript.
2. Authors are advised to reduce the number of keywords.
3. The studies for the adsorption of gas cyanogen chloride were reported by authors on the prepared impregnated activated carbon. Was the material was specifically designed to adsorb CK or is it capable to adsorb other gases too?
4. In characterization, the adsorption isotherm given does not seem to be good from the perspective of micropore range (Low pressure region). The authors are suggested to provide full adsorption isotherm in order to define the micro-porosity of the sample.
5. Did the authors carry out any study on disposal of these IAC beds after deactivation which are environment friendly?
Should be made better
Author Response
Our response to reviewer 1 comments:
We are very grateful to the reviewer for his time, efforts, and comments.
Below, please find our response to the comments:
Comment 1: The authors are advised to check the grammar of the manuscript.
Before submitting the article to “International Journal of Materials Science”, the article was submitted for full English language editing through AJE (American Journal Experts). AJE specializes in editing and correcting English in academic texts. The editing service that was ordered was a premium type, including “Correct grammar, phrasing, language, and punctuation” and “Premium language editing for natural and correct English by a US-trained expert in your field”. However, we will be happy to correct any further grammar mistakes that will be pointed out.
Nevertheless, following the reviewer's comment, we thoroughly reviewed the article, and indeed found a small number of required corrections. The corrections were made. In addition, phrasing improvements were made. The changes are marked (with the exception of minor changes in punctuation which were not marked).
Comment 2: Authors are advised to reduce the number of keywords.
Following the reviewer's comment, we reduced the number of keywords. The keywords list includes aging, activated carbon, cyanogen chloride, phosphate, impregnation, and adsorption.
Comment 3: The studies for the adsorption of gas cyanogen chloride were reported by authors on the prepared impregnated activated carbon. Was the material was specifically designed to adsorb CK or is it capable to adsorb other gases too?
The ASZMT carbon was not specifically designed to adsorb CK, but it is designed for use in both military and industrial respirator and collective filter applications. The ASZMT carbon can adsorb a variety of other gases, and CK is one of them. CK adsorption capacity is important since it is a chemical warfare agent.
The reason we used CK as the chemically adsorbed testing agent for aging is written in the manuscript at the end of the introduction: “As the protection capacity of the filter material against CK significantly decreases during aging, CK was used as the testing agent.”
Following the reviewer's question, we added in the introduction the following section:
The ASZMT carbon is effective for removing gases and/or vapors, including vapors of organic compounds, chlorine, hydrogen chloride, hydrogen cyanide, cyanogen chloride, sulfur dioxide, hydrogen sulfide, formaldehyde, and others.
Comment 4: In characterization, the adsorption isotherm given does not seem to be good from the perspective of micropore range (Low pressure region). The authors are suggested to provide full adsorption isotherm in order to define the micro-porosity of the sample.
The reason the isotherm presented in the article does not look good from the perspective of the micropore range is that the nitrogen adsorption isotherm does not allow the determination of the micropore size distribution. However, nevertheless, the nitrogen isotherm does make it possible to determine the micropore volume, the percentage of the volume of micropores from the total volume of the pores (% microporosity), as well as the general surface area, which includes the surface area of micropores. To determine the micropore size distribution, another method (CO2 adsorption-desorption isotherm) is required, which, unfortunately, is unavailable to us. At the same time, we would like to emphasize that the purpose of the isotherm was not to characterize the distribution of micropores in the carbon, but to examine whether the addition of phosphate changed the general surface area of the carbon, the volume of micropores, and the percentage of micropores out of the total volume of the pores. The adsorption isotherm of nitrogen answered these questions, and the shape of the isotherm remained type 1 even after adding phosphate. Therefore, we believe that although a CO2 adsorption isotherm could provide additional information, the nitrogen adsorption isotherm sufficiently provided essential information for the paper.
Comment 5: Did the authors carry out any study on disposal of these IAC beds after deactivation which are environment friendly?
The deactivated IAC, or used IAC, is sent by us to an official and regulated waste site. We are unaware of any other more environmentally friendly way to dispose of the IAC. However, it is worth noting that these days we are finishing a study aimed at reactivating IAC after aging. A paper about this research will be sent for publication very soon.
Reviewer 2 Report
Presented work provides important information on the ways for increasing of the activity in time to noxious substances of surface-modified active carbons, impregnated with Cu, Zn and Mo ions as well as TEDA (triethylenediamine). The main idea of the paper lays in addition of phosphates on the surface of the ACs to improve the adsorbent's resistance to aging over long periods of time (against the influence of high humidity), by retention of metals on their initial sites. The effect of phosphate addition on the protection efficiency of laboratory-impregnated carbon (IAC) and commercial carbon (ASZMT) after long aging periods (up to 6 months) was investigated in respect to protection properties towards cyanogen chloride (denoted as CK) and toluene using dynamic adsorption.
The results of the paper demonstrated the efficiency of the proposed method for the treatment of active carbons with phosphates, which provides significantly less degradation of the protective properties of aged adsorbents compared to the original ones in relation to CK.
The information provided in the paper is new and of scientific interest. The paper can be accepted in its current form; however there are several comments that may improve further research in this area:
1. It would be great to do the same set of experiments for “pure” unmodified carbon in the future for evaluation of the influence of only the porous structure.
2. It seems that temperature of regeneration of 80C can be rather low. Moreover, carbonaceous materials are usually outgassed and heated to over 150C before N2 adsorption–desorption measurements.
Minor editing of English language required
Author Response
Our response to reviewer 2 comments:
We are very grateful to the reviewer for his time, efforts, and comments.
Below, please find our response to the comments:
Comment 1: It would be great to do the same set of experiments for “pure” unmodified carbon in the future for evaluation of the influence of only the porous structure.
We agree with the reviewer that it will be great to examine his suggestion for pure, unmodified carbon in the future. As the reviewer pointed out, this comment refers to future work and does not require revisions in the current paper.
Comment 2: It seems that temperature of regeneration of 80 °C can be rather low. Moreover, carbonaceous materials are usually outgassed and heated to over 150 °C before N2 adsorption–desorption measurements.
We agree with the reviewer's comment that before the adsorption-desorption isotherm of carbonaceous materials, it is common to do outgassing at 150 °C. We performed the outgassing at 80 °C because, at 150 °C, there is a significant sublimation of TEDA from the IAC. This may possibly affect the measurement results. Furthermore, the sublimation speed of the TEDA from the IAC at high temperatures may vary for the different types of IACs. Therefore, we preferred to perform outgassing at 80 °C
Following the reviewer's comment, we added the following section in the experimental section:
The outgassing was performed at a temperature of 80 °C because, at higher temperatures, a significant sublimation of TEDA from the IAC may occur. Furthermore, the sublimation rate of the TEDA from the IAC at high temperatures may vary for the different types of IACs.
Comments on the Quality of English Language: Minor editing of English language required
Before submitting the article to “International Journal of Materials Science”, the article was submitted for full English language editing through AJE (American Journal Experts). AJE specializes in editing and correcting English in academic texts. The editing service that was ordered was a premium type, including “Correct grammar, phrasing, language, and punctuation” and “Premium language editing for natural and correct English by a US-trained expert in your field”. However, we will be happy to correct any further grammar mistakes that will be pointed out.
Nevertheless, following the reviewer's comment, we thoroughly reviewed the article, and indeed found a small number of required corrections. The corrections were made. In addition, phrasing improvements were made. The changes are marked (with the exception of minor changes in punctuation which were not marked).
Round 2
Reviewer 1 Report
Authors responded well; the manuscript is fine now.